# Wiring Up Vision: Minimizing Supervised Synaptic Updates Needed to Produce a Primate Ventral Stream

## Abstract

After training on large datasets, certain deep neural networks are surprisingly good models of the neural mechanisms of adult primate visual object recognition. Nevertheless, these models are poor models of the development of the visual system because they posit millions of sequential, precisely coordinated synaptic updates, each based on a labeled image. While ongoing research is pursuing the use of unsupervised proxies for labels, we here explore a complementary strategy of reducing the required number of supervised synaptic updates to produce an adult-like ventral visual stream (as judged by the match to V1, V2, V4, IT, and behavior). Such models might require less precise machinery and energy expenditure to coordinate these updates and would thus move us closer to viable neuroscientific hypotheses about how the visual system wires itself up. Relative to the current leading model of the adult ventral stream, we here demonstrate that the total number of supervised weight updates can be substantially reduced using three complementary strategies: First, we find that only 2% of supervised updates (epochs and images) are needed to achieve $\sim$80% of the match to adult ventral stream. Second, by improving the random distribution of synaptic connectivity, we find that 54% of the brain match can already be achieved "at birth" (i.e. no training at all). Third, we find that, by training only $\sim$5% of model synapses, we can still achieve nearly 80% of the match to the ventral stream. When these three strategies are applied in combination, we find that these new models achieve $\sim$80% of a fully trained model's match to the brain, while using two orders of magnitude fewer supervised *synaptic* updates. These results reflect first steps in modeling not just primate adult visual processing during inference, but also how the ventral visual stream might be "wired up" by evolution (a model's "birth" state) and by developmental learning (a model's updates based on visual experience).

## 1 Introduction

Particular artificial neural networks (ANNs) are the leading mechanistic models of visual processing in the primate visual ventral stream (Schrimpf et al., 2018; Kubilius et al., 2019; Dapello et al., 2020). After training on large-scale datasets such as ImageNet (Deng et al., 2009) by updating weights based on labeled images, internal representations of these ANNs partly match neural representations in the primate visual system from early visual cortex V1 through V2 and V4 to high-level IT (Yamins et al., 2014; Khaligh-Razavi & Kriegeskorte, 2014; Cadena et al., 2017; Tang et al., 2018; Schrimpf et al., 2018; Kubilius et al., 2019), and model object recognition behavior can partly account for primate object recognition behavior (Rajalingham et al., 2018; Schrimpf et al., 2018).

Recently, such models have been criticized due to how their learning departs from brain development (Marcus, 2004; Grossberg, 2020; Zador, 2019). For example, all the current top models of the primate ventral stream rely on trillions of supervised synaptic updates, i.e. the training of millions of parameters with millions of labeled examples over dozens of epochs. In biological systems, on the other hand, the at-birth synaptic wiring as encoded by the genome already provides structure that is sufficient for macaques to exhibit adult-like visual representations after a few months (Movshon & Kiorpes, 1988; Kiorpes & Movshon, 2004; Seibert, 2018), which restricts the amount of experience

dependent learning. Furthermore, different neuronal populations in cortical circuits undergo different plasticity mechanisms: neurons in supragranular and infragranular layers adapt more rapidly than those in layer 4 which receives inputs from lower areas (Diamond et al., 1994; Schoups et al., 2001), while current artificial synapses, on the other hand, all change under the same plasticity mechanism. **While current models provide a basic understanding of the neural mechanisms of adult ventral stream inference, can we start to build models that provide an understanding of how the ventral stream "wires itself up" – models of the initial state at birth and how it develops during postnatal life?**

**Related Work.**    Several papers have addressed related questions in machine learning: Distilled student networks can be trained on the outputs of a teacher network (Hinton et al., 2015; Cho & Hariharan, 2019; Tian et al., 2019), and, in pruning studies, networks with knocked out synapses perform reasonably well (Cheney et al., 2017; Morcos et al., 2018), demonstrating that models with many trained parameters can be compressed which is further supported by the convergence of training gradients onto a small subspace (Gur-Ari et al., 2018). Tian et al. (2020) show that a pre-trained encoder's fixed features can be used to train a thin decoder with performance close to full fine-tuning and recent theoretically-driven work has found that training only BatchNorm layers (Frankle et al., 2020) or determining the right parameters from a large pool of weights (Frankle et al., 2019; Ramanujan et al., 2019) can already achieve high classification accuracy. Unsupervised approaches are also starting to develop useful representations without requiring many labels by inferring internal labels such as clusters or representational similarity (Caron et al., 2018; Wu et al., 2018; Zhuang et al., 2019; Hénaff et al., 2019; Konkle & Alvarez, 2020; Zhuang et al., 2020). Many attempts are also being made to make the learning algorithms themselves more biologically plausible (e.g. Lillicrap et al., 2016; Scellier & Bengio, 2017; Pozzi et al., 2020) Nevertheless, all of these approaches require many synaptic updates in the form of labeled samples or precise machinery to determine the right set of weights. In this work, we take first steps of relating findings in machine learning to neuroscience and using such models to explore hypotheses about the product of evolution (a model's "birth state") while simultaneously reducing the number of supervised synaptic updates (a model's visual experience dependent development) without sacrificing high brain predictivity.

**Our contributions**    follow from a framework in which evolution endows the visual system with a well-chosen, yet still largely random "birth" pattern of synaptic connectivity (architecture + initialization), and developmental learning corresponds to training a fraction of the synaptic weights using very few supervised labels. We do not view the proposed changes as fully biological models of post-natal development, only that they more concretely correspond to biology than current models. Solving the entire problem of development all at once is too much for one study, but even partial improvements in this direction will likely be informative to further work. Specifically,

1. we build models with a fraction of supervised updates (training epochs and labeled images) that retain high similarity to the primate ventral visual stream (quantified by a brain predictivity score from benchmarks on Brain-Score (Schrimpf et al., 2018)),
2. we improve the "at-birth" synaptic connectivity to achieve reasonable brain predictivity with no training at all,
3. we propose a thin, "critical training" technique which reduces the number of trained synapses while maintaining high brain predictivity,
4. we combine these three techniques to build models with two orders of magnitude fewer supervised synaptic updates but high brain predictivity relative to a fully trained model

Code and pre-trained models will be available through GitHub.

## 2    MODELING PRIMATE VISION

We evaluate all models on a suite of ventral stream benchmarks in Brain-Score (Schrimpf et al., 2018), and we base the new models presented here on the CORnet-S architecture as this is currently the most accurate model of adult primate visual processing  (Kubilius et al., 2019).

**Brain-Score benchmarks.**    To obtain quantified scores for brain-likeness, we use a thorough set of benchmarks from Brain-Score (Schrimpf et al., 2018). To keep scores comparable, we only included those neural benchmarks from Brain-Score (Schrimpf et al., 2018) with the same predictivity

metric. All benchmarks feed the same images to a candidate model that were used for primate experiments while "recording" activations or measuring behavioral outputs. Specifically, the V1 and V2 benchmarks present 315 images of naturalistic textures and compare model representations to primate single-unit recordings from Freeman et al. (2013) (102 V1 and 103 V2 neurons); the V4 and IT benchmarks present 2,560 naturalistic images and compare models to primate Utah array recordings from Majaj et al. (2015) (88 V4 and 168 IT electrodes). A linear regression is fit from model to primate representations in response to 90% of the images and its prediction score on the held-out 10% of images is evaluated with Pearson correlation, cross-validated 10 times. The behavioral benchmark presents 240 images and compares model to primate behavioral responses from Rajalingham et al. (2018). A logistic classifier is fit on models' penultimate representations on 2,160 separate labeled images. The classifier is then used to estimate probabilities for 240 held-out images. Per-image confusion patterns between model and primate are compared with a Pearson correlation. All benchmark scores are normalized by the respective ceiling. We primarily report the average brain predictivity score as the mean of V1, V2, V4, IT, and behavioral scores.

We note that the Brain-Score benchmarks in this study are based on a limited number of data and thus present a possible limitation. Nonetheless, they are the most extensive set of primate ventral stream benchmarks that is currently available and the scores seem to generalize to new experiments (Kubilius et al., 2019).

Brain-Score provides separate sets of data as public benchmarks which we use to determine the type of distribution in Section 4, and the layer-to-region commitments of reference models.

**CORnet-S.** The current best model on the Brain-Score benchmarks is CORnet-S (Kubilius et al., 2019), a shallow recurrent model which anatomically commits to ventral stream regions. CORnet-S has four computational areas, analogous to the ventral visual areas V1, V2, V4, and IT, and a linear decoder that maps from neurons in the model's last visual area to its behavioral choices. The recurrent circuitry (Figure 3B) uses up- and down-sampling convolutions to process features and is identical in each of the models visual areas (except for $V1_{COR}$), but varies by the total number of neurons in each area.

We base all models developed here on the CORnet-S architecture and use the same hyper-parameters as proposed in (Kubilius et al., 2019). Representations are read out at the end of anatomically corresponding areas.

## 3 High scores in brain predictivity can be achieved with few supervised updates

We evaluated the brain predictivity scores of CORnet-S variants that were trained with fewer epochs and images. Models are trained with an initial learning rate of 0.1, divided by 10 when loss did not improve over 3 epochs, and stopping after three decrements.

Figure 1 shows model scores on neural and behavioral Brain-Score measures, relative to a model trained for 43 epochs on all 1.28M labeled ImageNet images. In Panel A, we compare the average score over the five brain measures of various models to the number of supervised updates that each model was trained with, defined as the number of labeled images times the number of epochs. While a fully trained model reaches an average score of .42 after 55,040,000 supervised updates (43 epochs $\times$ 1.28M images), a model with only 100,000 updates already achieves 50% of that score, and 1,000,000 updates increase brain predictivity scores to 76%. Models are close to convergence score after 10,000,000 supervised updates with performance nearly equal to full training (97%). Scores grow logarithmically with an approximate 5% score increase for every order of magnitude more supervised updates.

Figures 1B and C show individual neural and behavioral scores of models trained with fewer training epochs or labeled images independently. Early to mid visual representations (V1, V2, and V4 scores) are especially closely met with only few supervised updates, reaching 50% of the final trained model in fractions of the first epoch (Figure 1B). After only one full iteration over the training set, V1, V2, and V4 scores are close to their final score (all >80%) while IT requires two epochs to reach a comparable level. Behavioral scores take slightly longer to converge (>80% after 7 epochs).

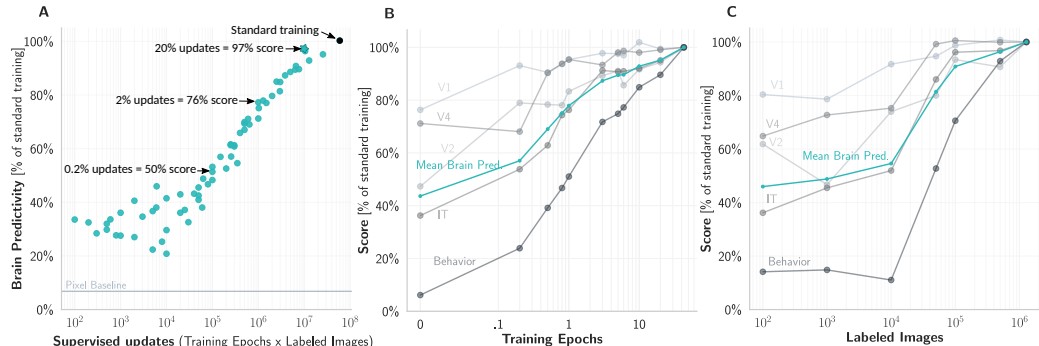

Figure 1: **High scores in brain predictivity can be achieved with few supervised updates** (log x-axes). **A** Average brain predictivity of models trained with a range of supervised updates (epochs × images). Each dot is a different hypothesis of how the ventral visual stream might have developed and shows the adult brain-likeness score that is achieved by that model. Fairly brain-like representations are already realized with few supervised updates, relative to a fully trained model (black dot; see also Figure 6). Standardly trained CORnet-S is set to be 100% score in brain predictivity on the benchmarks (0.42 absolute). A "pixels" baseline achieves 7% brain predictivity score (0.03 absolute). **B** Individual brain predictivity scores over epochs. Models start to approximate the primate ventral stream with few training epochs. Lower visual areas (V1, V2) are approximated earlier in training. **C** Like B, but number of training images instead of epochs. Few images are sufficient to approximate especially early visual areas.

Similarly, when training until convergence with fractions of the 1.28M total images, 50,000 images are sufficient to obtain high neural scores (80% of full training in V1, V2, V4, IT). Behavioral scores again require more training: half the standard number of labeled images is needed to surpass 80%.

Concretely relating supervised updates to primate ventral stream development, Seibert (2018) establishes that no more than ∼4 months – or 10 million seconds – of waking visual experience is needed to reach adult-level primate IT cortex (as assessed by its capability to support adult level object recognition). From this estimate, we can compute how many supervised updates per second different models in Figure 1A would require (assuming those updates are evenly distributed over the 10 million seconds). For instance, the fully trained model's 55 million supervised updates translate to 5.5 updates every second, whereas the model with 1 million updates and 76% relative brain predictivity translates to one labeled image update every 10 seconds which appears more plausible given the upper limit of 2-3 saccades per second in humans (Yarbus, 1967; Gibaldi & Sabatini, 2020).

## 4 "AT-BIRTH" SYNAPTIC CONNECTIVITY YIELDS REASONABLE BRAIN PREDICTIVITY WITH NO TRAINING AT ALL

If few supervised updates can get model representations fairly close to a fully trained model (Figure 1), how close are the initial representations without any training? In relation to biology and following the introduced framework of treating all consecutive training as developmental learning, these "at-birth" synaptic connections would result from information encoded in the genome as a product of evolution.

Due to the genome's capacity bottleneck, it is infeasible to precisely encode every synapse. Primary visual cortex alone contains ∼1.4E8 neurons per hemisphere (Leuba & Kraftsik, 1994), ∼1E3 synapses per neuron, each requiring ∼37 bits per synapse (Zador, 2019). Thus, without any clever rules, specifying the connections in only one hemisphere of V1 could require up to ∼5.2E12 bits – orders of magnitude more than the entire genome's 1GB = 8E9 bits (Zador, 2019). Sampling synaptic weights from reasonably compressed distributions on the other hand places only little memory requirements on genetic encoding while potentially yielding reasonably useful initial weights. Current machine learning techniques for initializing weights, such as Kaiming Normal (He et al., 2015), sample from a Gaussian distribution with $\mu = 0$ and $\sigma = \sqrt{2}/N$ where $N$ is the number of incoming connections per layer.

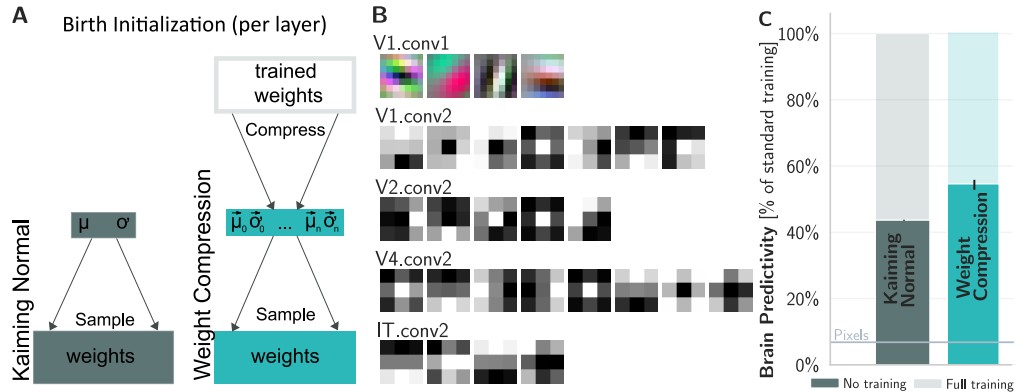

Figure 2: **"At-birth" synaptic connectivity yields reasonable scores in brain predictivity. A** Specifying the initial weight distribution: Kaiming Normal (*KN*, He et al., 2015) samples from a generic Gaussian. Weight Compression (*WC*) compresses trained weights into low-parameter clustered distributions that weights can be initialized from. **B** Visualization of *WC* compressed parameters: Gabor filters for first layer and cluster centers for following layers with kernel size $> 3$. The cluster centers capture an intuitive variety of kernel types. **C** "At-birth" representations with *WC* achieve 54% score in brain predictivity of a fully trained model, with no training at all. Scores after training remain virtually unchanged.

To improve on Kaiming Normal initialization, we explored multi-dimensional distributions as a more expressive alternative. These distributions only require a small number of parameters, but unlike current generic initializers, we explicitly specify them for each layer. To determine the right parameterization, we compress a trained model's weights into clusters which we then sample from (*Weight Compression, WC*).

More specifically, for all convolutional layers except the first, we cluster the kernel weights and later sample from the clusters. To capture the relative importance of clusters we fit a normal distribution to the cluster frequency over kernels. In batch normalization layers, we fit one normal distribution each to the weights and biases. For the first convolutional layer only, we employ a Gabor prior on the weights following studies in V1 (Hubel & Wiesel, 1962; Jones & Palmer, 1987) (supplement).

Model interpretability studies (Zeiler & Fergus, 2013; Olah et al., 2020; Cammarata et al., 2020) classify model weights, comparable to *WC*'s representation. Visualizing the weight compressions from trained CORnet-S weights (Figure 2B), we find that the first layer's Gabor filters qualitatively align with an analysis by Cammarata et al. (2020). Cluster centers seem to represent an intuitive division of channel types with opposite types in every layer.

While intuitively sampling weights from a compression of trained weights should somewhat recover reasonable brain predictivity scores, not every implementation satisfies both high brain predictivity and sufficient compression for the genome bottleneck (Appendix B.3).

Applying *WC* to CORnet-S, we first obtain a compressed and clustered set of parameters, from which we sample entirely new weights to yield a new model CORnet-S$_{WC}$. This model is *not trained at all* and we only evaluate the goodness of its initial wiring on the suite of Brain-Score benchmarks. Strikingly, we find that even without any training, CORnet-S$_{WC}$ achieves $54 \pm 1.5\%$ of the brain predictivity score relative to a fully-trained model (Figure 2), representing a 12 percent point improvement ($n = 10$ seeds; permutation test $p < 1E - 5$) over the Kaiming Normal initialized model with a score of $43 \pm 1.7\%$. Early ventral stream regions V1 and V2 are predicted especially well with no loss in score but we note that these two benchmarks are less well predicted by the trained model to begin with. V4 scores also approximate those of a trained model relatively well (75%). The major drop occurs in the IT and especially behavioral scores where CORnet-S$_{WC}$ only reaches 39% and 6% of the trained model's score respectively. Similarly, a trained linear decoder on CORnet-S$_{WC}$'s IT representations only reaches 5% of a trained model's ImageNet top-1 accuracy.

*Weight Compression* explores the hypothesis that evolution may have discovered an initialization strategy with improved at-birth representations (relative to current initializations). *WC* is most likely

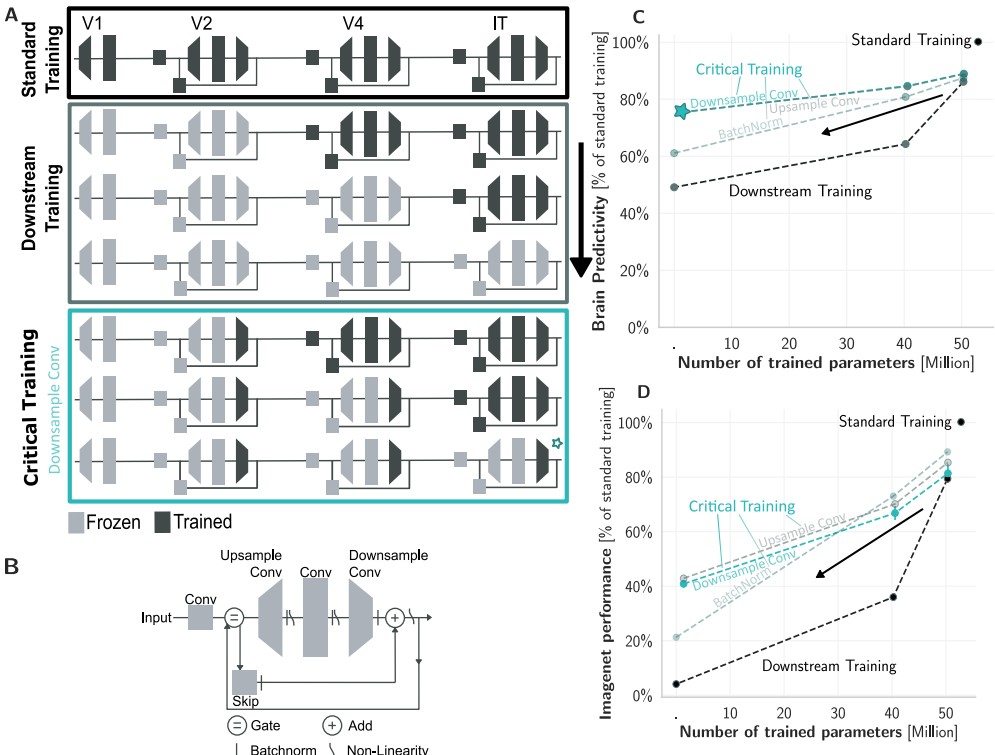

Figure 3: **Training only critical layers reduces the number of updated synapses while maintaining high brain predictivity**. **A** We could naively reduce the parameters of a fully-trained model by freezing layers from the bottom up, training only the top layers (*"Downstream Training DT"*; gray box). We instead propose *Critical Training (CT)* which only trains down-sampling layers (blue box). **B** CORnet-S circuitry. *CT* only trains critical layers, leaving the rest frozen. **C** Naively reducing parameters from standard training (black dot, top right) quickly deteriorates brain predictivity scores (*DT*, gray line) whereas *Critical Training* reduces parameters while retaining high scores (blue line, *CT*). **D** Like C, but measuring ImageNet score. *CT* retains nearly half the score with a fraction of parameters.

not how evolution found the at-birth synaptic connections, but shows that with nearly identical capacity, an alternative initialization distribution leads to networks that are more brain-like in their adult state – revealing a new space of possibilities (hypotheses) that should be considered.

## 5 TRAINING THIN DOWN-SAMPLING LAYERS REDUCES THE NUMBER OF UPDATED SYNAPSES WHILE MAINTAINING HIGH BRAIN PREDICTIVITY

While improved "at-birth" connectivity can reach 54% of a fully-trained model's score, additional visual-experience dependent updates appear necessary to reach higher predictivities. With standard training, each iteration simultaneously updates all of the millions of synaptic weights in the neural network, which may be difficult to implement in the brain. Alternatively, learning could take place preferentially in specific components. Cortical circuits are heterogeneous and different neuronal populations undergo distinct plasticity mechanisms. For example, neurons in supragranular and infragranular layers adapt more rapidly than those in layer 4, where inputs from lower areas arrive, as observed in rat somatosensory cortex (Diamond et al., 1994) and primate V1 (Schoups et al., 2001).

As a proof-of-principle that training a reduced set of layers can retain high performance, we propose a novel thin training technique, which we term *Critical Training (CT; Figure 3A). CT* updates only the weights in critical layers, instead of updating every single model synapse. In CORnet-S, each of the blocks has one down-sampling layer to produce an area's final representation (Figure 3B). We explore successive variants of applying *CT* up to a block in the architecture and then training the

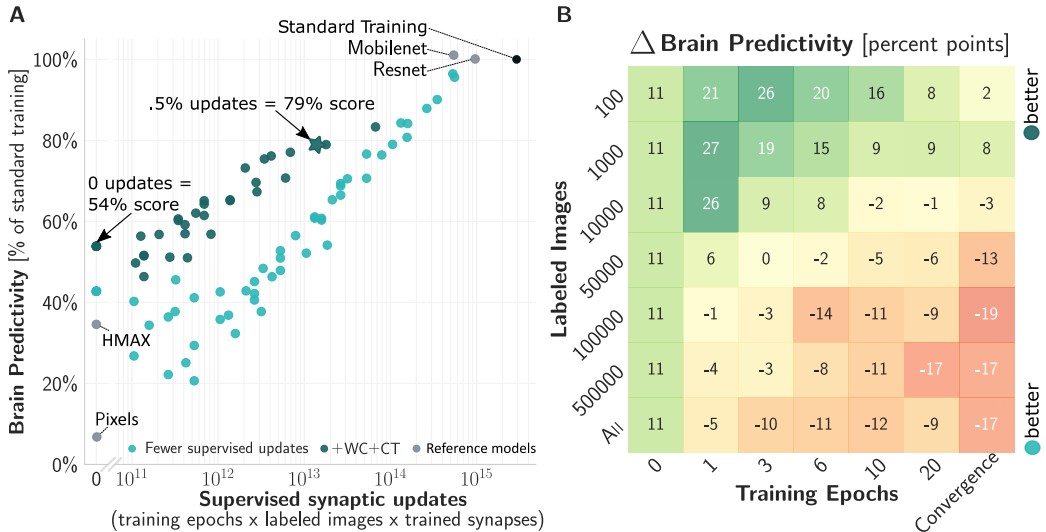

Figure 4: **High brain predictivity can be achieved with few supervised *synaptic* updates through a combination of training reductions** (log x-axis). **A** By reducing updates with a combination of fewer supervised updates (Figure 1), improved initialization *WC* (Figure 2), and training only down-sampling layers *CT* (Figure 3), the resulting models (dark blue dots; fewer supervised updates alone in light blue) maintain high brain predictivity scores while requiring only a fraction of supervised synaptic updates compared to standard CORnet-S (black dot, top right). **B** Comparison between *WC*-initialized models trained with *CT* versus standardly initialized models training all weights, when varying training epochs and labeled images. Colors represent their percent point difference in brain predictivity scores. *WC+CT* improve performance in regimes with few epochs and images.

following blocks, e.g. freezing V1, V2, V4 with critical training of the respective down-sampling layers and additional IT training. The final *CT* ventral stream model is almost completely frozen and only the synapses generating each cortical area's output are trained.

We compared *Critical Training* to a naive approach of reducing the trained parameters by freezing entire model blocks, for instance keeping the V1 and V2 blocks fixed while training V4 and IT blocks. We term this block-wise freezing and training approach *Downstream Training (DT)*. Compared to standard back-propagation training all the weights, both *CT* and *DT* reduce the number of trained parameters (Figure 3C). However, while the average score with *DT* (gray) already drops below 65% with over a quarter of trained parameters remaining, *CT* (blue) maintains over 75% with only 1.4 out of 52.8 million parameters trained. The choice of which critical layers to train also matters: training the connecting layers between regions – i.e. the last (down-sampling, default) or first (up-sampling) layer – retains most of the performance whereas training layers such as BatchNorm performs worse.

By reducing the number of trained parameters, *Critical Training* also yields engineering benefits in training time with a 30% reduction in the time per epoch at over 80% of the score in brain predictivity and more than 40% of the ImageNet score. The training time reduction is less drastic than the parameter reduction because most gradients are still computed for early down-sampling layers.

## 6 HIGH BRAIN PREDICTIVITY CAN BE ACHIEVED WITH A RELATIVELY SMALL NUMBER OF SUPERVISED SYNAPTIC UPDATES

All three training reduction methods independently minimize the number of supervised synaptic updates required to reach a reasonably high brain predictivity score. Reducing the number of supervised updates minimizes required updates by a smaller number of epochs and images (Section 3); *Weight Compression (WC)* improves the at-birth synaptic connectivity for high initial scores with no training at all (Section 4); and *Critical Training (CT)* reduces the number of synapses that are updated during training (Section 5). We now combine these three methods to build novel models that

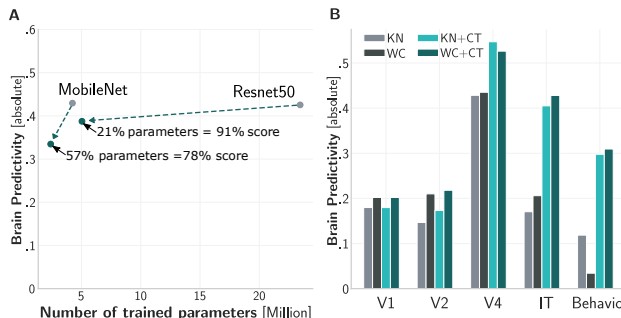

Figure 5: **Dissecting training reductions. A** Transfer to other networks. We sample from *WC* initializations determined on CORnet-S, followed by *Critical Training* of only down-sampling layers. **B** Absolute scores on individual benchmarks of combinations of initialization (*KN/WC*, Figure 2), and with critical training (*CT*, Figure 3) techniques.

only require a small number of supervised synaptic updates to reasonably capture the mechanisms of adult ventral visual stream processing and object recognition behavior.

Figure 4A shows the average brain predictivity of a range of models with varying numbers of supervised synaptic updates relative to a standard trained CORnet-S (black dot, 3,000 trillion supervised synaptic updates). With a reduced number of supervised updates (training epochs and labeled images) but standard initialization and training all weights (light blue dots), models require 5.2 trillion updates to achieve >50% of the score of a fully trained model and about 100 trillion updates to reach 80% brain score. Adding *WC+CT* (dark blue dots), the corresponding model already reaches 53% at birth with 0 supervised synaptic updates. At 0.5% the updates of a fully trained model (14 trillion vs. 3,000 trillion), models then reach 79% of the score (☆ model with modeling choices marked in Figures 1 to 3). Reference models (gray dots) MobileNet (Howard et al., 2017) and ResNet (He et al., 2016) obtain high scores, but also require many supervised synaptic updates. HMAX (Riesenhuber & Poggio, 1999) is fully specified with no updates but lacks in score.

We next examined interactions between the methods by comparing models initialized with *WC* and trained with *CT* to models with standard initialization and training all weights, when both are trained with fewer epochs and images. Figure 4B shows the percent point difference between the two model families. *WC+CT* yield strong benefits (green numbers) in a regime with few supervised updates, improving by up to 27 percent points when training for only 1 epoch on 1,000 images. With many updates on the other hand, *WC+CT* is actually less advantageous than standard training (red numbers): with all 43 epochs and 1.28M images, the score reduces by 17 percent points. *WC+CT* therefore most positively interact with a small budget of supervised updates, which we focus on in this work.

## 7    DISSECTING TRAINING REDUCTIONS

We asked whether the developed techniques would generalize to architectures other than the CORnet-S architecture that they were based on, perhaps a novel way to construct model taxonomies. We therefore applied *Weight Compression (WC)* and *Critical Training (CT)* to ResNet-50 (He et al., 2016) and MobileNet (Howard et al., 2017) architectures, both high-performing models on Brain-Score. We used *WC* distributions determined on CORnet-S, i.e. we tested *transfer* without re-fitting. *WC+CT* maintain 91% of the score in ResNet despite an almost 80% reduction in parameters. When applied to MobileNet, the average score drops by 22% and parameters are reduced less strongly (43%). This difference in retaining the score could be due to MobileNet already being very compressed, or having a less similar architecture.

With most analyses so far comparing an average score, we dissected the relative contributions of *WC* and *CT* to individual benchmarks (Figure 5B). We compared *KN* to *WC* initialization, as well as resulting models after critical training (*KN+CT* and *WC+CT*). *WC* initialization improves most over *KN* in early visual regions V1 and V2, while additional training with *CT* is most beneficial in mid- to high-level visual cortex V4 and IT, as well as the behavioral benchmark.

## 8    DISCUSSION

We developed a range of models with neural and behavioral scores approaching those of the current leading model of the adult ventral visual stream, while requiring only a fraction of supervised

synaptic updates. These models were built by complementarily 1) reducing the number of supervised updates, i.e. training epochs and labeled images; 2) improving the "at birth" distribution of synaptic connectivity; and 3) training only critical synapses at the end of each model area. The techniques and resulting models proposed here are first steps to more closely modeling not just adult primate visual processing, but also exploring the underlying mechanisms of evolution and developmental learning.

These proof-of-principle demonstrations are far from accounting for the rich information encoded in the genome or the developmental learning that together result in adult mechanisms of visual processing. We here started from CORnet-S, which is the current leading model of the adult ventral stream, but does not fully predict all brain measurements. The architecture we based our techniques on might therefore be flawed. We verified favorable transfer to models with similar architectures such as ResNet, but generalization to an already compressed MobileNet was limited (Figure 5A).

Relating to genomic mechanisms, the proposed techniques should generalize to other domains such as auditory processing. With the capacity bottleneck in the genome, mechanisms for wiring up would likely be shared between similar systems. The fact that early visual areas converge earlier during training (Figure 1) and are better predicted than higher areas by *WC* initialization is consistent with developmental studies of the primate ventral stream. In humans, behaviors that rely on low-level spatial and temporal processing of visual inputs reach adult-like performance considerably earlier than complex visual behaviors that rely on higher cortical regions, such as face perception (Ellemberg et al., 1999; Grill-Spector et al., 2008).

A critical component in more closely modeling primate development is to reduce the dependence on labels altogether. Recent unsupervised approaches are starting to rival the classification performance of supervised models (Caron et al., 2018; Hénaff et al., 2019; Zhuang et al., 2020) and combining them with the advances presented here could further reduce the number of synaptic updates. More precise biological measurements are required to quantify the number of (parallel) experience-dependent updates. Current unsupervised techniques however still require back-propagation which is routinely criticized as non-biological, among others due to the propagation of gradients (Grossberg, 1987; Whittington & Bogacz, 2019; Hunsberger, 2017). Local learning rules might alleviate these concerns and with critical training (Figure 3), it could be sufficient to learn in only a subset of layers.

The changes to model initialization and training presented here serve as a proof-of-principle that models can be changed to more closely align with primate development by reducing training steps with labeled images and improving initialization. It is also possible to achieve high brain predictivity when training only down-sampling layers, but all these models are still far from the actual biological mechanisms. We expect future work in this direction to further close the gap with improved evolutionarily encoded wiring mechanisms and developmental learning rules.

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

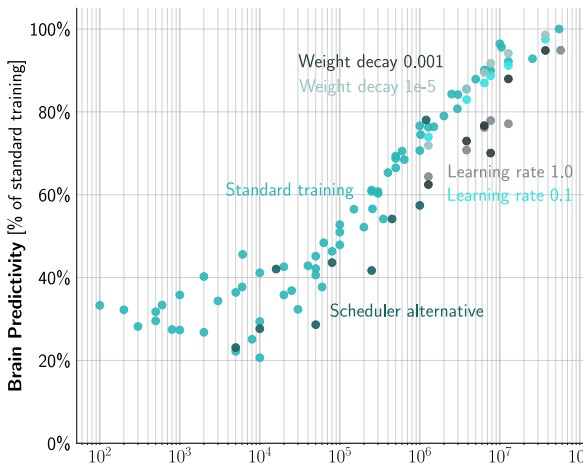

Figure 6: **Findings generalize across hyperparameters.** Notations like in Figure 1. To test whether fast learning could be achieved by purely changing hyperparameters, we evaluated different regularizations (weight decay 0.001 and 0.00001 vs. 0.0001), learning rates (initial 1.0 and 0.01 vs. 0.1 in the main manuscript) and learning rate schedules ("Scheduler alternative" decreases learning rate more agressively). Trends are qualitatively similar across all choices, and quantitatively nearly optimal with respect to fast convergence to high brain predictivity for the hyperparameters used in the manuscript.

## A    BENCHMARK DETAILS

We use the benchmarks as implemented in `www.github.com/brain-score/brain-score` at commit 96b0711, and convert base models to brain models with `www.github.com/brain-score/model-tools` at commit 2f778c6. Images were presented at 4 degrees without aperture for the V1 and V2 benchmarks and at 8 degrees for the V4, IT, and behavior benchmarks. Models committed to an input size of 8 degrees visual angle.

## B    WEIGHT COMPRESSION DETAILS

For all convolutional layers except the first, we cluster kernel weights in a layer using the k-means algorithm (Fix & Hodges, 1951). The number of clusters is determined using elbow (Thorndike, 1953). To capture the relative importance of clusters we fit a normal distribution $\mathcal{N}_f$ for each cluster with $\mu_f$ as the cluster frequency over kernels and $\sigma_f$ as the frequency standard deviation. To sample weights for a kernel, we first sample a cluster distribution $i \sim \mathcal{N}_f$ per kernel and then obtain channel weights by sampling from a Gaussian with $\vec{\mu}_i$ as the cluster center and the standard deviation $\vec{\sigma}_i$ of clustered weights. In batch normalization layers, we fit one normal distribution each to the weights and biases.

### B.1    COMPRESSING THE FIRST LAYER WITH A GABOR PRIOR

The weight compression approach we use in Section 4 is based on different initialization techniques, applied to different layers. For the very first layer of size $7 \times 7$ we found a Gabor filter most effective following studies in V1 (Hubel & Wiesel, 1962; Jones & Palmer, 1987). To generate the Gabor kernels we fit trained channel weights to a Gabor function

$$G_{\theta,f,\phi,n_x,n_y,C}(x,y) = \frac{1}{2\pi\sigma_x\sigma_y} \exp\left[-0.5(x_{rot}^2/\sigma_x^2 + y_{rot}^2/\sigma_y^2)\right] \cos\left(2\pi f + \phi\right) C \qquad (1)$$

where

$$\begin{aligned} x_{rot} &= x\cos(\theta) + y\sin(\theta) \\ y_{rot} &= -x\sin(\theta) + y\cos(\theta) \end{aligned} \qquad (2)$$

$$\begin{aligned} \sigma_x &= \frac{n_x}{f} \\ \sigma_y &= \frac{n_y}{f} \end{aligned} \qquad (3)$$

$x_{rot}$ and $y_{rot}$ are the orthogonal and parallel orientations relative to the grating, $\theta$ is the angle of the grating orientation, $f$ is the spatial frequency of the grating, $\phi$ is the phase of the grating relative to the Gaussian envelope, $\sigma_x$ and $\sigma_y$ are the standard deviations of the Gaussian envelope orthogonal and parallel to the grating, which can be defined as multiples ($n_x$ and $n_y$) of the inverse of the grating

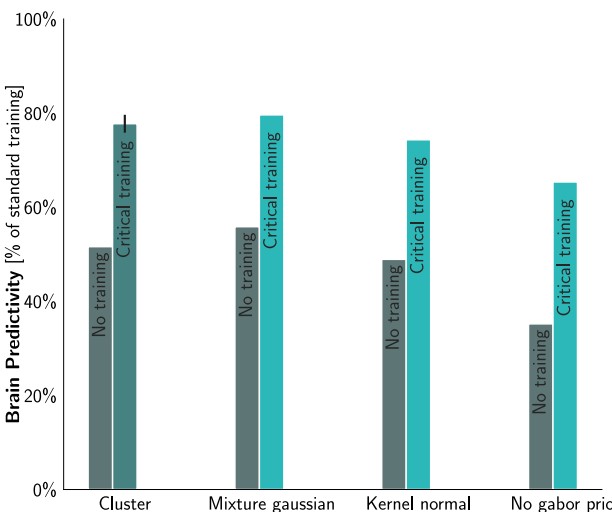

Figure 7: **Alternative weight compression methods** Comparison of different initializations that compress weights, "at birth" i.e. without any training (gray) and after training critical layers (shades of blue) for 6 epochs. Our best clustering-based approach *WC* achieved similar results as the *Mixture Gaussian* approach (∼3 percent points mean difference) but leads to more diverse clusters. Performance drops when solely sampling weights from kernel based normal distributions (*Kernel normal*) and additionally disabling the Gabor prior (*No Gabor prior*)

frequency and $C$ is a scaling factor.

The function is fit per channel, which leads to a set of Gabor parameter for each of the 3 RGB channels. We then fit a multidimensional mixture of Gaussians to the combination of all filter parameter per kernel, resulting in a kernel parameter set. For the three RGB input channels in the first layer and the 8 Gabor parameters we therefore fit to $3 \times 8 = 27$ parameters. We evaluate the best number of components (number of distinct Gaussian distributions) based on the Bayesian Information Criterion (Schwarz, 1978) and use 4 components for the first layer of CORnet-S. To generate new kernels we sample a kernel parameter set from this mixture distribution and apply them to the described Gabor function that spans the weight values.

## B.2 COMPRESSING BATCHNORM LAYERS

In addition to convolutional layers, models consist of several Batchnorm layers, which contain a learnable bias and weight term. To initialize these terms, we fit a normal distribution per weight and bias vector of the trained values and sample from this distribution. Note that BatchNorm layers contain running average means and standard deviations for normalization purposes, which are applied at validation time. Those terms are set to zero when no training has happened, but cause score changes once the model has processed the dataset. During training the mean and standard deviation of the current batch are used instead.

## B.3 ALTERNATIVE APPROACHES

We have explored a variety of weight compression methods applied to different layers and evaluate their performance "at birth" without training and when trained with critical training.

Figure 7 shows brain predictivities of several alternative compression methods implemented as follows:

- **WC** Weight compression approach with clustering as described in Section 4, using a Gabor prior approach for the first layer, noisy cluster sampling for convolutional layers and fitted normal distributions for Batchnorm layers (4,166 parameters for CORnet-S).

- **Mixture Gaussian** Instead of sampling weights from cluster centers, this approach uses multidimensional distributions for convolutional layers with kernel size > 1. We fit a mixture Gaussian distribution per layer to the weights of a channel over all kernels. To sample a new kernel, we sample individual channels from this distribution. For convolutional layers with a kernel size of 1 we draw weights from a normal distribution adjusted per kernel as described in the next item (428,114 parameters for CORnet-S).

- **Kernel normal** All weights are sampled based on normal distributions. We fit mean and standard deviation to the weights of one trained kernel and resample a new kernel from this

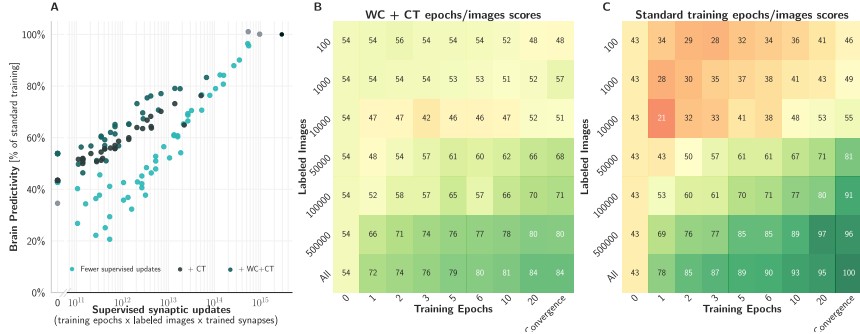

Figure 8: **Detailed analysis of *WC+CT*. A** When reducing the number of supervised synaptic updates, adding critical training (dark grey) and adding weight compression initialization (dark blue) both improve scores in brain predictivity at the same number of supervised synaptic updates, in comparison to a model with standard initialization and training all weights (bright blue). **B** Brain predictivities for the *WC+CT* model when trained with a range of epochs and labeled images. **C** Same as B, but for a standardly initialized (*KN*) model training all weights.

distribution. We do this separately for every kernel to generate a whole layer. This approach is similar to the BatchNorm sampling method where we compress BatchNorm weight and bias terms instead of kernels (433,735 parameters for CORnet-S).

- **No Gabor prior** To evaluate the importance of the Gabor prior we use the **Kernel normal** model and apply the same normal distribution approach to layer one instead of Gabor sampling. Performance drops by 13 percent points without training, and by 9 percent points after critical training. (20,026 parameters for CORnet-S)

## C  *WC* INITIALIZED AND *CT* TRAINED MODEL ANALYSIS

Our best model *WC+CT* benefits from a combination of improved initialization through weight compression, and critical training. Figure 8A shows models with standard initialization and training all weights, but with fewer supervised updates (cf. Figure 1), models that only train down-sampling layers (*CT*), and models that combine critical training with weight compression (*WC+CT*). A model initialized with weight compression achieves (only *WC*) 54% brain predictivity score with 0 supervised synaptic updates. Figure 8B and C show detailed brain predictivity scores, relative to a fully trained model, for models initialized and trained with *WC+CT* (B) and models initialized with standard Kaiming Normal and training all weights (C) when trained with a range of epochs and labeled images. The specific benchmark scores when either training with all labeled images for a varying number of epochs (Figure 9A) or when training with fewer labeled images until convergence (Figure 9B) show the benchmarks of early visual achieve the best results, relative to a fully trained model. The V1 score is identical over all training states, since we do not train the V1 area.

## D  DISSECTING TRAINING REDUCTIONS – DETAILS

### D.1  TRANSFER TO RESNET AND MOBILENET

To show the generalization of our approach we applied the weight compression methods to a ResNet-50 (He et al., 2016) and a MobileNet (Howard et al., 2017) (version 1, multiplier 1.0, image size 224) architecture. We do not regenerate sampling distributions or clusters based on the new architectures trained weights, but used the CORnet-S based distributions to sample new weights for the different architectures. Since CORnet-S is inspired by ResNet modules, we applied our critical training approach by training all conv3 layers (equivalent down sampling layers) of ResNet50. For MobileNet we explored various layer mappings. When training only the very few layers that result in reduced feature size, which are implemented as depthwise separable convolutional layers and appear three times overall, performance performance dropped close to random. Those layers however are mapped

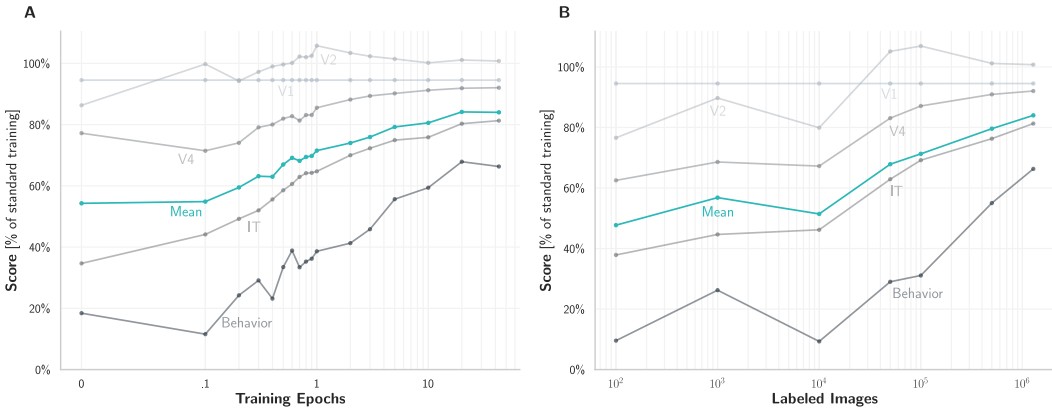

Figure 9: **Individual brain benchmark scores for *WC+CT* model A** Individual brain predictivitiy scores of *WC+CT* models trained with a range of epochs on all images. These models score especially high on V1, V2 and V4 already after one epoch in comparison to a model with standard initialization training all weights. IT and Behavior benchmarks continuously improve over later epochs as well but fall short of a fully trained model. **B** Like A, but with models trained until convergence on different numbers of labeled images, up to the full dataset of 1.28M images (rightmost points). As in A we see $> 80\%$ V1, V2 and V4 scores with only 100,000 images. For comparable IT and behavioral scores, more images are required.

to CORnet-S' conv2 layers due to their $3 \times 3$ kernels whereas critical training in CORnet-S trains conv3 down-sampling layers with a kernel size of $1 \times 1$. To transfer our critical training approach, we therefore additionally train the $1 \times 1$ MobileNet layers corresponding to conv3. This training version allows for more training but still reduces the amount of trained parameters by 43% while maintaining 78% of the original score. For both transfer methods we initialize the first layer using the Gabor method based on CORnet-S's mixture-of-Gaussian distribution. Since the Gabor function is scalable we can produce Gabor kernels of varying size. Furthermore we disable BatchNorm biases and weights in all transfer models by freezing them to default values. We found that transferring those distributions on new architectures harms brain predictivity scores. Nevertheless, the BatchNorm layers still normalize activations by applying the running average and standard deviation.

### D.2 COMPARISON OF TECHNIQUES TO REDUCE SUPERVISED SYNAPTIC UPDATES (FIG. 5B)

To analyse the relative contributions of *Weight Compression* and *Critical Training* we compare brain predictivitiy scores of different models in Figure 5B:

- **KN** A model initialized by standard *Kaiming Normal* initialization without training.
- WC A model initialized by our *Weight Compression* initialization, described in Section 4, without training.
- KN+CT The *KN*-initialized model trained with *Critical Training* until convergence, i.e. three downstream layers and the decoder are trained and all other layers remain unchanged.
- **WC+CT** The *WC*-initialized model with *Critical Training*. V1 scores do not change because weights in the V1 model area are all frozen.

## E TRAINING DETAILS

We used PyTorch 0.4.1 and trained the model using the ImageNet 2012 training set Deng et al. (2009). We used a batch size of 256 images and trained on a QuadroRTX6000 GPU until convergence. We start with a learning rate of 0.1 and decrease it four times by a factor of ten when training loss does not decrease over a period of three epochs. For optimization, we use Stochastic Gradient Descent with a weight decay 0.0001, momentum 0.9, and a cross-entropy loss between image labels and model logits. We trained all models with these settings except the standard Mobilenet, where we used the pretrained tensorflow model. Since the number of epochs for this model are not clearly

stated, we use the published value of 100 training epochs Howard et al. (2017). The training time of a full CORnet-S with standard Imagenet dataset for 43 epochs is ∼2.5 days. All variations with less weights/images/epochs trained in shorter time. Reference models trained for 4 days at most under the described settings. If not further specified, we show results of one training run. When showing error bars we used seeds 0, 42 and 94.

Code to reproduce our analyses from scratch, including the framework for weight compression and critical training, as well as pre-trained models, will be made available through GitHub.

