# OpenReview forum: "Wiring Up Vision: Minimizing Supervised Synaptic Updates Needed to Produce a Primate Ventral Stream"
_ICLR.cc/2021/Conference — Reject_

### Official Review · AnonReviewer3 · 2020-10-28
**Review for "Wiring Up Vision: Minimizing Supervised Synaptic Updates Needed to Produce a Primate Ventral Stream"**

**Rating:** 6
**Confidence:** 4

**Review:**

This paper presents an empirical study that elucidates potential mechanisms through which models of adult-like visual streams can "develop" from less specific/coarser model instantiations. In particular, the authors consider existing ventral stream models whose internal representations and behavior are most brain-like (amongst several other models) and probe how these fair in impoverished regimes of available labeled data and model plasticity (number of "trainable" synapses). They introduce a novel weight initialization mechanism, Weight Compression (WC), that allows their models to retain good performance even at the beginning of training, before any synaptic update. They also explore a particular methodology for fine-tuning, Critical Training (CT), that selectively updates parameters that seem to yield the most benefit. Finally, they explore these methods/algorithms' transfer performance from one ventral stream model (CORnet-S) to two additional models (ResNet-50 and MobileNet).

Pros:
The problem that the authors present is an interesting one and undoubtedly useful for many applications. Deep neural networks such as the CORnet-S, ResNet-50, and MobileNet are data-hungry, and obtaining labeled data is an expensive process (and perhaps even implausible in many cases). Techniques to condense these models in terms of parameters and alleviate the need for vast amounts of labeled data while maintaining desirable traits (such as brain-like representations) are important for the machine learning community. Though a bit far-fetched at this point, tracking the developmental trajectories of these neural networks can also have other scientific implications in the form of data-driven hypothesis testing.

The most exciting part of the study is the transfer experiment (from CORnet-S to ResNet and MobileNet). This seems like an interesting and novel way to construct model taxonomies. For instance, sampling from the CORnet-S weight clusters works well for ResNets potentially because these two models can be construed as "recurrent" in a way. MobileNets, on the other hand, are purely feedforward and thus are not significantly influenced by knowledge from the CORnet-S weights.

Moreover, the authors conduct a series of numerical experiments to identify "when" their proposed methods are most useful. The finding that WC+CT is more advantageous in regimes where data is scarce (as opposed to regimes where data is plenty) is not surprising but a good one to report. I say "not surprising" because WT distills knowledge from a fully trained model, and CT only updates a fraction of the parameters (updating more parameters would require more data to prevent overfitting).

Cons:
The authors take the analogy between "a developing visual system" and "training a model" a bit too far. They operate under the premise that visual circuitry develops purely via "supervised" learning. Is there conclusive evidence for this? It is also surprising that discussions of reinforcement learning mechanisms never feature, given that these are more biologically plausible.

The novelty (and utility; for ex: Fig 2b) of the proposed initialization technique is marginal. It is not articulated how their method (WC) overcomes the critiques they raise against Frankle et al. 2019. Moreover, claiming that WC achieves decent performance with "zero" synaptic updates is not fair. This seems to be closer to restoring pre-trained weights than to random initialization (like KN).

For CT, the authors choose "critical" layers to update. Is there a rationale (or a statistical metric) that justifies choosing these specific layers?

The WC kernel cluster center visualization analysis (Fig. 5c) seems out of place and poorly discussed. What can be gleaned from the 3x3 kernels shown here?

Minor:
By "supervised updates," the authors refer to the number of available labels and not the number of parameter updates that happen. This terminology is non-canonical.

Employing Gabor priors for the first convolutional layer: Doesn't orientation selectivity emerge in the primary visual areas from experience, rather than structurally hard-coded?

The authors allude to the possibility of using "local" learning rules on a subset of layers identified by CT. However, this is speculation from the point of view of the current manuscript. All the conclusions drawn are from "global" gradients.

Ambiguous sentence (Pg. 6, Sec 6): "Reducing the number of supervised updates minimizes required updates by a smaller number of epochs and images."

(Pg. 8) "synaptic updates primarily take place in higher cortical regions": Is there evidence for this?

Numerical imprecisions:
(i) The authors claim that the performance of CORnet-S_wc is 54% (relative to the fully trained model). However, in Fig 2b (mean) and Fig 3c (top) the markings seem to be closer to 50%?
(ii) (Fig. 4a) The performance of MobileNet seems to be slightly better than CORnet-S, which contradicts the initial claim that CORnet-S is currently the best available model of adult primate visual processing.

---

> ### Author Response · Authors · 2020-11-18
> **Initial Response 1/2**
>
> Thank you for your review and this great summary. To make the best constructive use of OpenReview, we wanted to send an initial response with the changes we are planning. Please let us know if those changes fully address your concerns with our study and if there are additional analyses that would be helpful to clarify any remaining questions.
>
> Re pros: We agree with the point about constructing model taxonomies in future work. Applying weight clustering in this way on a range of model architectures would be interesting to determine model similarities.
>
> We also share the intuition that WC+CT was expected to be most advantageous in a scarce data regime, that is exactly what we focused on in this work.
>
> Re cons: The models are indeed not yet accurate models of development. The analogies used in our study are the framework (p. 2) in which we lay out what the models should, in our view, aspire towards. We do not argue that the changes we have proposed are fully biological models of post-natal development, only that they more concretely correspond to biology than current models. By changing the models to be more inline with biological post-natal development while still achieving adult states that are very brain-like, we hope that the new models (i.e. computational hypotheses with all parameters fixed) will become more serious models of visual development.
>
> We see these concrete commitments of model stages to biological stages as essential so that we can concretely relate biological data with model predictions. They allow for a concrete tracking in time between model and biology from “birth” (instantiated architecture) via “development” (experience-dependent training updates) to “adulthood” (inference and adult-level learning).
>
> Following this framework, many aspects of current models’ development are non-biological. This study tackles the number of weight updates with a combination of reduced training, improved initialization, and training a subset of layers. Related works tackle for instance biologically plausible variants of back-propagation (e.g. Lillicrap et al. 2016, Scellier et al. 2017) or very recently self-supervision (Konkle et al. 2020, Zhuang et al. 2020). We will discuss these in more detail and also refer to Pozzi et al. 2020 for a concrete implementation of RL in this context. We will also make our partial approach of improving -- but not yet accomplishing -- models as hypotheses of biological development more clear.
>
> Weight Compression (WC) improves over Frankle et al. 2019 by specifying the weights in a compressed distribution instead of having to specify for every single weight whether it needs to be trained or not. This compression is biologically necessary due to the information bottleneck in the genome (section 4) where detailed specifications such as in Frankle et al. 2019 are inconsistent with the limited capacity of the genome. Weight Compression is an existence proof that such improved distributions can be found and in this case improve from 43% to 54% (Fig. 2B, see below for an update on the results reported in this figure). We don’t think evolution found these distributions by compressing learned weights; rather we view WC as a way of showing that there are better initial distributions that could have been optimized during evolution and encoded with the genome with little information.
>
> The critical layers in CT are those with the fewest weights to obtain the potential most minimal training. For comparison, training only the middle layer of the IT block would require training 38M of the 53M parameters (over 70%).
>
> We agree with the comment that 5C is out of place and we will move this subpanel to Fig. 2, where we present the results of the WC. By showing these kernel cluster centers, we wanted to connect our work with analytic interpretation studies (sec. 7, last paragraph). WC is a constructive approach to validating kernel clusters that could come out of post-hoc analyses.
>
> Could you clarify which papers’ definitions you are referring to with regards to the “supervised updates” terminology? We would like to use canonical terms and apologize for not being aware of existing definitions.
>
> Regarding emergence of orientation selectivity in the primary visual areas: In primates, not only orientation selectivity is present at birth, but the primary visual cortex already shows a topographical arrangement of orientation columns prior to visual experience (Wiesel and Hubel, 1974). The same is true for other mammalian species such as cat and ferret (for a review of the literature see Huberman, Feller and Chapman 2008).

---

> ### Author Response · Authors · 2020-11-18
> **Initial Response 2/2**
>
> Regarding the point about local learning: Indeed, we speculate about the possibility of local learning rules in the Discussion. We think this work here might enable such rules because Critical Training shows that training a subset of layers can be sufficient for reasonable accuracy and brain predictivity.
>
> We agree that the statement "synaptic updates primarily take place in higher cortical regions" is unsupported. We will update the corresponding paragraph in the manuscript. We maintain the conclusion that the fact that early visual areas in the model converge earlier is in agreement with neurodevelopmental studies. Behaviors that rely on low-level spatial and temporal processing of visual inputs reach adult-like performance very early - 4 years for temporal vision and 6 years for spatial vision (Ellemberget al 1999). On the other hand, more complex visual behaviors that rely on higher cortical regions, such as face perception, only fully develop at an age of around 16 years old (Grill-Spectoret al 2008).
>
> Re numerical imprecisions:
> (i) Indeed, there was a plotting error. We will fix this and have verified the correctness of results with additional runs and statistical analyses that confirm a score of 54+/-1.5% for WC, n=10 seeds; and improvement over 43+/-1.7% for KN, permutation test p < 1e-5.
> (ii) We only included Brain-Score benchmarks of one type (linear predictivity) in this study to keep results comparable across cortical regions. Brain-Score includes an additional IT benchmark measuring temporal correspondence of models which puts CORnet-S at the top overall. We will clarify this in the paper.

---

### Official Review · AnonReviewer4 · 2020-10-31
**Surprising reduction in number of weight updates needed to achieve a good Brain-Score**

**Rating:** 8
**Confidence:** 3

**Review:**

Summarize what the paper claims to contribute.
Previous work developed CORnet-S, a biologically inspired network that leads the Brain-Score benchmark of similarity with the primate ventral stream. A limitation of CORnet-S and other deep networks with high Brain-Scores is that they require many more weight updates than seem biologically feasible. In this paper, the number of weight updates used to train CORnet-S is reduced by two order of magnitude, while retaining a fairly high Brain-Score. This is done by combining three approaches, including reduced training, initialization of weights using compact distributions that describe trained weights, and updating only a minority of layers.

List strong and weak points of the paper.
Strong Points:
-	The paper addresses an important problem that has not been given much attention previously
-	The work builds on the state-of-the-art model in this domain
-	The three approaches to reducing updates are complementary and interesting in different ways; the second and third thought-provoking with respect to their biological relevance
-	The experiments and analysis are thorough
-	The paper is well written
-	The context of the work is clearly described and well referenced

Weak Points:
I wasn’t able to discern any substantial weaknesses.

Clearly state your recommendation (accept or reject) with one or two key reasons for this choice.
I recommend acceptance. The number of updates needed to learn realistic brain-like representations is a fair criticism of current models, and this paper demonstrates that this number can be greatly reduced, with moderate reduction in Brain-Score. I was surprised that it worked so well.

Ask questions you would like answered by the authors to help you clarify your understanding of the paper and provide the additional evidence you need to be confident in your assessment.
-	Is the third method (updating only down-sampling layers) meant to be biologically relevant? If so, can anything more specific be said about this, other than that different cortical layers learn at different rates?
-	Given that the brain does everything in parallel, why is the number of weight updates a better metric than the number of network updates?

Provide additional feedback with the aim to improve the paper.
-	Bottom of pg. 4: I think 37 bits / synapse (Zador, 2019) relates to specification of the target neuron rather than specification of the connection weight. So I’m not sure its obvious how this relates to the weight compression scheme. The target neurons are already fully specified in CORnet-S.
-	Pg. 5: “The training time reduction is less drastic than the parameter reduction because most gradients are still computed for early down-sampling layers (Discussion).” This seems not to have been revisited in the Discussion (which is fine, just delete “Discussion”).
-	Fig. 3: Did you experiment with just training the middle Conv layers (as opposed to upsample or downsample layers)?
-	Fig. 3: Why go to 0 trained parameters for downstream training, but minimum ~1M trained parameters for CT?
-	Fig. 4: On the color bar, presumably one of the labels should say “worse”.
-	Section B.1: How many Gaussian components were used, or how many parameters total? Or if different for each layer, what was the maximum across all layers?
-	Section B.3: I wasn’t clear on the numbers of parameters used in each approach.
-	D.1: How were CORnet-S clusters mapped to ResNet blocks? I thought different clusters were used in each layer. If not, maybe this could be highlighted in Section 4.

---

> ### Author Response · Authors · 2020-11-18
> **Initial Response**
>
> Thank you for your review and this great summary that well highlights why we are also excited about this work. To make the best constructive use of OpenReview, we wanted to send an initial response with the changes we are planning. Please let us know if those changes fully address your concerns with our study and if there are additional analyses that would be helpful to clarify any remaining questions.
>
> Regarding the update of only down-sampling layers: We do not know of any dataset with which we could make more precise biological commitments. As mentioned in the review, at this point Critical Training is primarily a proof-of-principle that different cortical learning rates (even in the extreme of no updates) can lead to useful representations.
>
> Regarding the weight updates metric: We report both the number of supervised updates (whole-model, parallel updates) as well as the number of supervised synaptic updates (per synapse) because we view both as relevant. To quantify experience-dependent updates and in lieu of more precise biological measurements (e.g. estimating the energy expenditure per synapse), we agree that supervised updates should be measured. We will clarify this point in the paper.
>
> Regarding bits/synapse: The 37 bits/synapse are indeed a possible under-estimate because more bits might be required to specify different synaptic strengths. Our main argument with this estimate was that it is infeasible to precisely encode “a pre-trained weight matrix” in the genome, necessitating either the learning of most weights (a point often made by deep learning advocates) or compressed initialization (such as WC proposed in this study).
>
> Pg. 5: Indeed, we will update this.
>
> Fig. 3 middle layers: We only experimented with layers with the fewest weights to obtain the potential most minimal training. For comparison, training the middle layer of the IT block would require training 38M of the 53M parameters (over 70%).
>
> Fig. 3 trained parameters: This is a result of the different training techniques: The most minimal point for Downstream Training freezes all parameters (see subpanel A, gray box, last row) whereas the most minimal point for Critical Training still updates a single layer per block (see subpanel A, cyan box, last row).
>
> Fig. 4: We will make the suggested improvement.
>
> Section B.1: We used 4 components for V1.conv1. Note that the Gabor prior is not used beyond V1.conv1. We will add these details to the text.
>
> Section B.3: WC uses 4,166 parameters, Mixture 428,114, Kernel Normal  433,735, No Gabor prior 20,026 to initialize the weights. We will add this to the text.
>
> Resnet Mapping: To initialize ResNet from the WC clusters, we mapped the ResNet architecture (blocks 0 - 4 ) to the CORnet-S architecture (blocks V1, V2, V4, IT) as follows 0 → V1, 1 → V2, 2 → V4, 3 → V4, 4 → IT. Since ResNet blocks have more layers but no recurrence, the CORnet-S layers are mapped repeatedly. Based on the mapping layers we initialized weights from the CORnet-S cluster centers.

---

### Official Review · AnonReviewer1 · 2020-11-03
**Interesting yet unconvincing ideas about modeling the primate visual system with DNNs**

**Rating:** 3
**Confidence:** 4

**Review:**

The study starts from the fact that DNNs have been around and popular for a while for modeling the visual system, but that they are not realistic because they are trained via supervised learning approaches with a very large number of parameters and that this is not a feasible model of the development in the visual system.

In general, although the manuscript presents some interesting ideas, it makes many assumptions without providing clear bases for these assumptions (e.g. compressing the weights of a pretrained network to sample new weights is posed as a realistic approximation of the infant visual brain) and lacks a theoretical foundation for the claims and experiments that are presented. The authors acknowledge that this study is intended as a proof of principle, but given the arbitrary nature of the choices made, I do not see the added significant value of the results.

While DNNs are indeed commonly used as models of the primate visual system, in my view, the current study is addressing a somewhat inconsequential problem. This is because to the best of my knowledge, no neuroscientist is claiming that a deep neural network is a complete and accurate model of the (development of the) primate visual system. Furthermore, it is well-known and acknowledged that deep neural networks are not biologically plausible models of (how learning occurs in) the brain. They are currently one of the best computational tools to use to study the sensory (and especially the visual) nervous systems, and that is all that they are. It is not clearly explained why it is necessary to claim that the learning in these models and the development of the brain has to be similar for them to be good models of vision. Of course, we should thrive for better and more accurate models of the brain, but in my view the current study does not serve to this goal.

In section 4 authors describe an initialization protocol for the network weights which involve compressing a trained model’s weights into clusters and then sampling from these clusters. What is not clear is why the authors assume that this can be a valid model of the infant visual system. At this point their approach sounds like arbitrarily selecting a set of criteria to make the networks perform worse than fully trained networks, and then training them. I could be missing something, but I do not see the relevance or necessity of an approach such as the presented one. A main concern is that no theoretical basis has been established in the paper besides some superficial ideas. For instance, why would an infant brain be made up of a DNN with connections whose weights are initialized with the method authors came up with?

Much of the methodological details are only included in the appendix. I found it rather odd to not find any information about, for example, the proposed weight initialization method in the paper.

It is not clear to me what is presented in Figure 1 and why. Why are the authors showing how models from another paper trains?

Another concern is that nowhere in the results seems to be a test for significance. The improvements of the results could be a coincidence, since the results are heavily dependent on one experiment.

---

> ### Author Response · Authors · 2020-11-18
> **Initial Response 1/2**
>
> Thank you for your review. To make the best constructive use of OpenReview, we wanted to send an initial response with the changes we are planning.
> The central argument of this review is that current artificial neural networks are not good models of visual system development -- a point that we completely agree with and is the motivation of the explorations of different types of models that we undertook to test in this paper.  The main specific criticism seems to be that we did not justify the biological assumptions that underpin our choices of the specific models that we chose to test.   We agree that our choices should be better explained and justified and we plan to make those changes as outlined below.   However, we hope that the reviewer will agree that, because no other image-computable models are even close to explaining the adult ventral visual stream, exploration of the minimal assumptions needed to achieve models that are at least as brain-explanatory as the current deep CNN models is a contribution to the field that might deserve visibility in ICLR.
>
> ### No current ANN is a complete and accurate model of the (development of the) primate visual system
> We absolutely agree that no current artificial neural network (ANN) is a complete model of the primate visual system and its development — that is precisely our motivation for trying to find alternative ANNs that are more brain-like. In this study,  we focus on model changes that might be more inline with biological post-natal visual development.  Current deep ANN models of the visual system are criticized for being non-biological due to their reliance on an excessive amount of supervised weight updates (e.g. Grossberg 1988, 2020, Marcus 2004). While other studies have focused on other aspects such as investigating biologically plausible implementations of supervised learning (e.g. Lillicrap et al. 2016, Scellier et al. 2017) or very recently self-supervision (Konkle et al. 2020, Zhuang et al. 2020), we here studied the amount of training (both in labeled images and in synaptic updates) required to achieve adult states that are still very brain like and propose initialization and training procedures that reduce the amount of training required.   We do not argue that the changes we have proposed are fully biological models of post-natal development, only that they are more biological than the current models.  The idea is that solving the entire development problem all at once is too much for one study, but that even partial improvements in this direction will be informative to further work.  We will make the framing of this partial approach clear in the updated paper version.
>
> ### ANNs as computational hypotheses of primate object recognition
> We disagree with R1 in the sense that, in our view, fully-trained ANN models  are not just tools, but they are computational hypotheses (i.e. approximate models) of brain processing.  Specifically, each such model (all parameters fixed) can be aligned and tested at all levels of the adult ventral visual stream and any failed predictions of neural responses at any level falsifies that model and can be used to guide the building of new, improved models. To date, certain, fully trained, ANN models  are the most accurate hypotheses of ventral stream processing.  Here, by exploring alternative, more biologically-plausible ways of discovering such ANN models, we hope that the new models will become more serious models of  visual development.  The first test of any such model is its match to the adult visual data, so our approach pivots off of that measure.   Whether more biologically plausible models of development will lead to benefits for Machine Learning is an open research question that we do not engage with in our study.  But existing studies already suggest that closer modeling of biology will have additional benefits, e.g. increased generalization (Kubilius et al. 2019) and robustness (Dapello et al. 2020). The “worst” outcome of such modeling efforts is a better model of biological development which, in our view, is an exciting research direction in itself.
> These benefits are of course still mostly in the future and our study does not solve them all. But we are taking first steps towards them by showing that the number of supervised synaptic updates in brain models can be more closely aligned with biological development without a severe decrease in match of the “adult” model to the adult brain.

---

> ### Author Response · Authors · 2020-11-18
> **Initial Response 2/2**
>
> ### Sampling initial weights from an improved distribution
> We do not claim that weight compression is how evolution found the at-birth synaptic connections. In our study, we are simply using alternative models to explore the hypothesis that evolution may have discovered an initialization strategy that leads to higher behavioral performance and higher match to the ventral stream than current initialization distributions.   Because of the genomic bottleneck, the initialization cannot  specify all the developed weights, so there is a trade-off between capacity of information in the genome and at-birth goodness of neural representations. Our results reveal that with nearly identical capacity, an alternative initialization distribution (relative to that the one typically used) leads to networks that are more brain-like in their adult state.  Our study’s contribution is not to prove that the brain uses such a strategy to set up its initial synaptic distribution, but to reveal a new space of possibilities (hypotheses) that should be considered -- the hallmark of any good modeling study.  We will clarify this in the paper.
>
> ### Figure 1 presents models from different developmental trajectories
> Each dot (model with a certain training) in Figure 1 is a different hypothesis of how the ventral visual stream might have developed.  Each dot corresponds to a model architecture trained for a certain amount of time (epochs and labeled images) and the adult brain-likeness that is achieved by that model. To the best of our knowledge, this is the first work that presents a multitude of neural and behavioral scores over these different models and shows that: early visual representations of some models are very adult-like (i.e. matched to the brain data) with very little supervised experience, and that high-level visual representations of all models currently require more supervised experience to achieve comparable levels of adult brain match.   We plan to update the description of this figure to make this more clear in the text.
>
> ### Tests for significance
> For Figure 2, we have updated the comparison between Kaiming Normal (KN) and Weight Compression (WC) with a larger number of seeds and performed statistical tests to show that our method significantly improves brain predictivity (43+/-1.7% vs 54+/-1.5% for KN and WC relative brain predictivity respectively, n=10 seeds; permutation test p < 1e-5). Due to the amount of different models trained, we only have one seed per model in Figure 3 but the differences in magnitude and the consistency of results leave no doubt for the improvements of the Critical Training (CT) over the Downstream Training.

---

### Official Review · AnonReviewer2 · 2020-11-04
**How much can we rely on BrainScore's metric in studies like this one?**

**Rating:** 6
**Confidence:** 5

**Review:**

Summary
-------
The paper is about ANN being best-known models of developed primate visual systems. However this fact does not yet mean that the way those systems are trained is also similar. This distinction and a step towards answering this question is the main motivation of this work. The authors demonstrate a set of ideas that while drastically reducing the number of updates maintain high Brain Predictability according to the BrainScore. The significance of this result in my opinion largely depends on how well we can map those observations and methods to biological meaning and knowledge on how primate brains are trained (see the discussion point below).


Critique, Questions, Discussion
-------------------------------
(1) How good the "match" between the brain and DCNN is in the first place? For example, if we measure the match in terms of correlation (between responses, or predictions, any metric would work in the context of this question), then 80% of corr=1.0 would be very impressive and significant, while 80% of corr=0.2 (being corr=0.16) could well fall under the noise and while being significant numerically, does not give us the opportunity to say that we have captured 80% of the match between the artificial system and the ventral stream (because what we have actually captured is 80% of corr=0.2, which might as well be almost nothing).

(2) "squirrels to jump from tree to tree within months of birth", "macaques to exhibit adult-like visual representations after months" -- hoe many synaptic updates happen during those months? Do we know? Maybe it is also in trillions? In which case this portion of the argument would fall apart. Emphasis on "supervised" would probably still survive.

(3) "a child would need to ask one question every second of her life to receive a comparable volume of labeled data" -- are they not? I would say children get even more data if by "question" we will mean not only verbal questions and answers, but also answers that are tactile ("how this will feel on touch?"), auditory ("what does this object sound like"), visual prediction ("will this thing now move to the right or to the left?"), etc. Seen like this I would say that children receive tons of supervised data and "one per second" is an underestimation.

(4) How does the "match" vary depending on random initialization? Is it consistently 54% or is there a substantial +/-?

(5) How do we know the "true zero" in terms of the "match"? What would be a model (function? maybe a constant function?) that clearly has zero "match"? If we now take this function and run it through your pipeline to get the match%, would the result be indeed 0% or something else? Maybe 54% is the "true zero" and not 0%.

(6) Why sampling from CORnet-S-based clusters of parameters is a good way of modeling "at-birth" situation? Compared to 54% achieved with this methods, what would be the match% if the network would initialized with vanilla Kaiming Normal? Uniform?


Recommendation and justification
--------------------------------
My main concern is with the interpretation of the meaning of this work. BrainScore's metric is a very approximate proxy that weakly reflects the match between models of vision. In this work, however, this metric is taken as a "gold standard" and it is assumed that achieving, for example 50% of BrainScore of 0.42 is something biologically meaningful. An ablation experiment that would demonstrate that achieving these 50% (or other numbers presented in the paper) is a non-trivial event which can only happen if the model is indeed becoming more "brain-like" would go a long way in making the case of this work strong. I suspect, however, that such an ablation study will show that there are ways to achieve high% of BrainScore using models that are completely dissimilar to the brain. I currently evaluate this submission as borderline, and am looking forward to authors' views on the concerns I have outlined above: do these indeed matter and affect the claims of this work (and how should we see them if that's the case), or are these concern largely irrelevant (and why we can ignore them if that's the case?).


Additional remarks
------------------
Arguably missing references on modeling of the ventral stream with ANNs: https://www.nature.com/articles/s42003-018-0110-y, https://www.jneurosci.org/content/35/27/10005


UPDATE - Nov 30
-----------------------
After looking at the revised version of the manuscript I am still concerned that the claims made in the abstract (and implied in the main text of the paper) about the match of ANNs to the brain are misleading the reader into assigning greater biological significance to the reported result than it actually holds. While the authors made slight modifications in the text and added a few sentences commenting on the issue, these changes did not constitute a change would make the reader "extremely aware that when you say "80% match" you don't mean "80% match to the brain", but "80% match to the score"". I find that a softer claim that would explicitly acknowledge that 5% of "synaptic" updates explain 80% of the predictivity score and not 80% of the match to the brain would make this work more scientifically precise and thus more valuable. I am keeping my original assessment of this paper as being borderline.

---

> ### Author Response · Authors · 2020-11-18
> **Initial Response**
>
> Thank you for your review. To make the best constructive use of OpenReview, we wanted to send an initial response with the changes we are planning. Please let us know if those changes fully address your concerns with our study and if there are additional analyses that would be helpful to clarify any remaining question.
>
> (1) The standard-trained CORnet-S achieves a score of 0.42 relative to an estimated ceiling (second-to-last line on page 3). For comparison, a pixel baseline only achieves 0.03, i.e. 7% on the normalized plots in the paper. We have this baseline in Figure 2 and Figure 4 but will also add it to Figure 1 for reference.
>
> (2+3) This is a great question and as far as we know the exact number of synaptic updates is unknown. There is an upper limit based on the update rate of long-term synaptic plasticity. Another upper limit we use in the paper is that humans saccade only 2-3 times per second, so the number of new images we receive is limited. In some sense, we are trying to motivate more accurate biological estimates so that models can be falsified -- currently,  such estimates are rather vague giving a lot of leeway to models. We will make these upper limits more clear in the introduction. We will also remove the quote about children following your comment because we agree that self-supervision could serve to obtain labels for each saccade.
>
> (4) We plot standard deviation respectively over multiple runs in Figure 2B. We have run more seeds to confirm statistical differences following Reviewer 1’s remark showing that Weight Compression scores significantly higher than vanilla Kaiming Normal (43+/-1.7% for KN vs 54+/-1.5% for WC, n=10 seeds; permutation test p < 1e-5).
>
> (5) The primary “zero” baseline we are using are the pixel values (pixels in Figure 2 and 4) which achieve only 7% of the standard-trained model. Therefore, the 54% achieved by WC without training far improves over this baseline.
>
> (6) Vanilla Kaiming Normal initialization achieves 43% (Figure 2B), we will add this to the text.
>
> Regarding the use of the Brain-Score suite of benchmarks:
> * Brain-like by definition is to match and predict observed data (be it neural, behavioral, or anatomical).
> * To the best of our knowledge, Brain-Score is the gold standard for comparing models of the ventral stream on an integrative set of neural and behavioral benchmarks across the regions V1, V2, V4, and IT that support core object recognition behavior.
> * Many analyses to confirm the validity of the benchmarks have been reported in previous papers:
>     * Different architectural choices lead to considerable spread in the scores (https://www.biorxiv.org/content/10.1101/407007v2, Fig. 1)
>     * Model scores generalize to new images and new primates/recordings (https://papers.nips.cc/paper/9441-brain-like-object-recognition-with-high-performing-shallow-recurrent-anns, Fig. 2)
>     * Models with a better V1 match also better match behavior in terms of adversarial robustness (https://papers.nips.cc/paper/2020/hash/98b17f068d5d9b7668e19fb8ae470841-Abstract.html)
> * Figure 1 in our study also addresses one version of an ablation experiment: reducing the model’s developmental process leads to a considerable spread in the scores (reducing them as far as 20% of the original model’s score), but perhaps also not as severe of an immediate reduction with fewer updates as many in the field would have thought.
> * In Figure 5A, we also made sure that the distributions found for CORnet-S through WC and the critical training (CT) generalizes to Resnet50 and MobileNet architectures. (Reviewer 3 makes an interesting remark about using these techniques to create model taxonomies)
> * Please let us know if there is a particular additional analysis that you had in mind and we would be happy to run it.

---

> > ### Comment · AnonReviewer2 · 2020-11-19
> > **Digging deeper into Question 1**
> >
> > Thank you for the comments. With the exception of Q1 they are a satisfactory response to my criticism.
> > Now regarding Q1 - it was meant as a somewhat deeper question than just "what kind of score is considered good". Let me try to have another go at it.
> >
> > Step 1: Imagine you've built a model that has a match of 1.0 according to BrainScore. Would you then conclude that the representation employed by this model is 100% identical to the representation that is employed by the brain? I would assume you would not, because the fact that it can perfectly predict brain responses from ANN activity does not yet mean that those systems have similar representation.
> >
> > (Let me note that I appreciate that this is not you, but rather on the creators of BrainScore. However using using BrainScore as your metric _is_ on you, which allows me to pick at you here :) )
> >
> > Step 2: Now let's hypothetically assume that scoring 1.0 on BrainScore corresponds to being 3% identical in terms of representations (a rather low number, I agree, but let me show where this thought experiment leads). CORnet-S achieves 0.747. Multiplied by 3% it means (using our hypothetical assumption) that CORnet-S's representation is 0.747 * 3% = 2.241% identical to brain's representation. And with 5% synaptic updates, as you show, we capture 80% of that, putting us at 1.7928% identical. Given all the noisiness and approximate nature of biological readings would you still say that the difference between 1.7928% and 2.241% has biological meaning?
> >
> > Step 3: A sentence like "we find that only 2% of supervised updates (epochs and images) are needed to achieve ~80% of the match to adult ventral stream" makes a reader believe that since you achieved 80% of score then those 2% capture "a lot" of similarity between ANNs and brains, while in reality it only means that it captures 80% of the _score_, but how much of the similarity between the ANNs and brains it is would fully depend on how biologically meaningful is the score itself.
> >
> > Another way of putting it: a reader should be made extremely aware that when you say "80% match" you don't mean "80% match to the brain", but "80% match to the score". If this is said explicitly and repeatedly the reader has a chance to understand that the actual significance of this result depends on how good the score is as a metric of similarity between ANNs and brains.
> >
> > A way out of this problem would be somehow quantify what does BrainScore=1.0 actually mean in terms of closeness of representations of the two systems, but that is a highly non-trivial task.
> >
> > Even a more drastic thought experiment would be if we assume (just as an experiment) that 1.0 BrainScore is achievable even with absolutely no similarity between representations of an ANN and the brain. In this case capturing 80% of the score would not tell anything about the match between representations...
> > This remains my main concern.

---

> > > ### Author Response · Authors · 2020-11-21
> > > **Follow-up Response**
> > >
> > > Thank you for your quick response and the opportunity to discuss this concern! Your clarification was very helpful -- if we understand correctly, there are two related overarching concerns:
> > >
> > >  1. **Wording**: 80% match on Brain-Score is not necessarily the same as 80% match to the brain due to the limited number of datasets. The current text does not make this clear. We completely agree with this point and will update the language in the paper to make this more explicit.
> > >
> > >  2. **The “goodness” of Brain-Score**: It is unclear to what extent the current benchmarks on Brain-Score are aligned with the “real” representational similarity to the brain.
> > > We acknowledge this criticism but would also like to push back on some positions. This point is perhaps a bit more philosophical, so we will try to explain in more detail where we are coming from.
> > >
> > > 2.1 The “real” representational similarity to the brain can be operationalized as the match of a model on all benchmarks and datasets that could ever possibly exist. These benchmarks could employ very different metrics, stimuli, or recording techniques than the ones in use today. Either way, this (extremely large) set of benchmarks contains the current (very small) set of benchmarks on Brain-Score, so there has to be some alignment. The generalization of scores to new stimuli and primate recordings as shown in Fig. 2, https://papers.nips.cc/paper/9441-brain-like-object-recognition-with-high-performing-shallow-recurrent-anns, makes us hopeful that this alignment is fairly substantial. If a specific ANN can perfectly predict brain responses, it thus has to employ similar representations to the brain.
> > >
> > > 2.2 Quantifying what Brain-Score=1.0 on a subset of benchmarks actually means with respect to the “complete” set of benchmarks (2.1) is impossible to address until we have collected all the data we could ever hope for. At that point, the benchmarks are perfectly aligned with the “real” representational similarity by definition. In practical terms, we can approximate this alignment with held-out benchmarks (as mentioned in 2.1) and we see no other way forward than to work with the benchmarks at hand, while adding more and more benchmarks that break the models. As far as we are aware, the benchmarks in Brain-Score are the most extensive set of primate ventral stream benchmarks that is currently readily available.
> > >
> > > 2.3 It is thus in our view impossible that a high score on even the current Brain-Score benchmarks can be achieved with absolutely no “real” representational similarity. In the limit of infinite data outlined above, a score of 1.0 would require an exact copy of the brain.
> > >
> > > 2.4 To offer a practical suggestion that could perhaps serve to connect the functional scores to anatomy: we could test how well the model would score if its mapping of layers to regions were jumbled up. That is, what would the score be if we used the V1 layer to predict IT and the IT layer to predict V1? Our prediction would be that this anatomical mismatch should lead to decreased scores -- likely still well above zero because V1 and IT also predict each other in the brain, but it might at least give a little more confidence in a qualitative match between model and brain processing.
> > >
> > > Either way, we appreciate your constructive criticism and will definitely work the points from this discussion (that we are happy to continue!) into the final paper.

---

### Author Response · Authors · 2020-11-25
**Combined Response 1/2**

We sent individual responses to all reviewers to potentially be able to discuss directly through OpenReview, but also wanted to post an overall response to address common criticisms.

## ANNs as models of development

R1 and R3 raised concerns over the analogy of artificial (deep) neural networks to the development of the primate visual system. R1 for instance states “no neuroscientist is claiming that a deep neural network is a complete and accurate model of the (development of the) primate visual system” and R3 adds that our study “operate[s] under the premise that visual circuitry develops purely via "supervised" learning.”

We absolutely agree that no current artificial neural network (ANN) is a complete model of the primate visual system and its development — that is precisely our motivation for trying to find alternative ANNs that are more brain-like. In this study, to better compare ANNs to the development of the primate ventral stream, we make the explicit commitment of ANNs initialization to "birth-state" and ANN training to experience-dependent learning. We then focus on changes to these two stages that may be more in line with biological post-natal development. Current deep ANN models of the visual system are criticized for being non-biological due to their reliance on an excessive amount of supervised weight updates (e.g. Grossberg 1988, 2020, Marcus 2004). While other studies have focused on other aspects such as investigating biologically plausible implementations of supervised learning (e.g. Lillicrap et al. 2016, Scellier et al. 2017) or very recently self-supervision (Konkle et al. 2020, Zhuang et al. 2020), we here studied the amount of training (both in labeled images and in synaptic updates) required to achieve adult states that are still very brain like and propose initialization and training procedures that reduce the amount of training required. We do not argue that the changes we have proposed are fully biological models of post-natal development, only that they more concretely correspond to biology than current models. While, solving the entire development problem all at once is too much for one study, here we take the first steps  in this direction, laying the ground for future work.
We have updated the paper and made the framing of this partial approach of improving -- but not yet accomplishing -- models as hypotheses of biological development more clear. We have also expanded the related work section to discuss approaches addressing other ANN shortcomings of biological learning in more detail.


## Brain-Score as a metric for evaluating alignment of models with the primate ventral visual stream

R2 points out a limitation in the Brain-Score benchmarks that were used to quantify the match between models and primate ventral stream. Since they are based on a limited number of datasets, the scores might not generalize to new datasets and the wording did not make this clear.

We have updated the text to make the use of the benchmarks in this study as limited proxies more explicit, but also to point out results that support the generalization of even the current set of Brain-Score V1, V2, V4, IT, and behavior benchmarks. Most importantly https://papers.nips.cc/paper/9441-brain-like-object-recognition-with-high-performing-shallow-recurrent-anns, Fig. 2, showed that model scores generalize to new images and new primates/recordings. Figure 1 in our study also addresses one version of an ablation experiment: reducing the model’s developmental process leads to a considerable spread in the scores (reducing them as far as 20% of the original model’s score), but perhaps also not as severe of an immediate reduction with fewer updates as many in the field would have thought. As R4 nicely pointed out, the techniques themselves seem to generalize well to reasonably similar architectures (Figure 5A).

---

> ### Author Response · Authors · 2020-11-25
> **Combined Response 2/2**
>
>
> ## Biological rationales of proposed methods
>
> R1, R2, R3 have questioned the use of weight compression (WC) as a model of primate post-natal development. Since weight compression bases distributional clusters on trained weights, a similar biological mechanism is difficult to imagine. It is therefore also unclear under what measures WC can be fairly compared to standard Kaiming Normal initialization.
>
> We do not claim that weight compression is how evolution found the at-birth synaptic connections. In our study, we are simply using alternative models to explore the hypothesis that evolution may have discovered an initialization strategy that leads to higher behavioral performance and higher match to the ventral stream than current initialization distributions.   Because of the genomic bottleneck, the initialization cannot  specify all the developed weights, so there is a trade-off between capacity of information in the genome and at-birth goodness of neural representations. Our results reveal that with nearly identical capacity, an alternative initialization distribution (relative to that the one typically used) leads to networks that are more brain-like prior to training. Specifying the weights in a compressed distribution is a significant improvement over Frankle et al. 2019 where it needs to be specified for every single weight whether it needs to be trained or not. Our study’s contribution is not to prove that the brain uses such a strategy to set up its initial synaptic distribution, but to reveal a new space of possibilities (hypotheses) that should be considered -- the hallmark of any good modeling study.  We have clarified this in the paper.
>
> We have also run additional statistical analyses following R1’s remarks to confirm that WC (54+/-1.5%) indeed improves over standard Kaiming Normal initialization (43+/-1.7%, n=10 seeds; permutation test p<1e-5).
>
> Please see the individual reviewer threads for more detailed responses.

---

### Decision · Program_Chairs · 2021-01-07
**Final Decision**

**Decision:**

Reject

**Comment:**

This paper received 2 borderline accepts, 1 accept, and 1 reject.

This paper was discussed on the forum and no consensus was reached. The two reviewers who rated the paper as borderline accept emphasized that the biological claims are overblown, that the intellectual contributions (the initialization scheme and partial training) are incremental from a statistical learning perspective, and that the potential applications for the future (like alternate learning rules) are too speculative. I agree with both of these reviewers (and the negative reviewer) that the biological rationale is problematic and the approach is not credible as a model of biology. It is not evaluated as a computer vision model either. And I completely agree with the point raised by several reviewers that there is simply no data about how many synaptic updates to target. Hence, statements regarding % of total synaptic updates and % of brain matches seem empty without a precise target. For all these reasons, I recommend this paper be rejected.